# Spermine-Mediated Tolerance to Selenium Toxicity in Wheat (*Triticum aestivum* L.) Depends on Endogenous Nitric Oxide Synthesis

**DOI:** 10.3390/antiox10111835

**Published:** 2021-11-19

**Authors:** Md. Mahadi Hasan, Basmah M. Alharbi, Haifa Abdulaziz Sakit Alhaithloul, Awatif M. Abdulmajeed, Suliman Mohammed Alghanem, Amina A. M. Al-Mushhin, Mohammad Shah Jahan, Francisco J. Corpas, Xiang-Wen Fang, Mona H. Soliman

**Affiliations:** 1State Key Laboratory of Grassland Agro-ecosystems, School of Life Sciences, Lanzhou University, Lanzhou 730000, China; hasanmahadikau@gmail.com; 2Biology Department, Faculty of Science, Tabuk University, Tabuk 71491, Saudi Arabia; b.alharbi@ut.edu.sa (B.M.A.); s-alghanem@ut.edu.sa (S.M.A.); 3Biology Department, College of Science, Jouf University, Sakaka 2014, Saudi Arabia; haifasakit@ju.edu.sa; 4Biology Department, Faculty of Science, University of Tabuk, Umluj 46429, Saudi Arabia; dr.aabdulmajeeda@gmail.com; 5Department of Biology, College of Sciences and Humanities in AlKharj, Prince Sattam Bin Abdulaziz University, AlKharj 11942, Saudi Arabia; a.almushhin@psau.edu.sa; 6Key Laboratory of Southern Vegetable Crop Genetic Improvement in Ministry of Agriculture, College of Horticulture, Nanjing Agricultural University, Nanjing 210095, China; shahjahansau@gmail.com; 7Department of Horticulture, Sher-e-Bangla Agricultural University, Dhaka 1207, Bangladesh; 8Group of Antioxidants, Free Radicals and Nitric Oxide in Biotechnology, Food and Agriculture, Estación Experimental del Zaidín, Department of Biochemistry, Cell and Molecular Biology of Plants, 18008 Granada, Spain; javier.corpas@eez.csic.es; 9Botany and Microbiology Department, Faculty of Science, Cairo University, Giza 12613, Egypt; monahsh3344@gmail.com; 10Biology Department, Faculty of Science, Taibah University, Yanbu 46429, Saudi Arabia

**Keywords:** antioxidant enzymes, gene expression, glyoxalase systems, oxidative stress, reactive oxygen species (ROS)

## Abstract

Excess selenium (Se) causes toxicity, and nitric oxide (NO)’s function in spermine (Spm)-induced tolerance to Se stress is unknown. Using wheat plants exposed to 1 mM sodium selenate—alone or in combination with either 1 mM Spm, 0.1 mM NO donor sodium nitroprusside (SNP) or 0.1 mM NO scavenger cPTIO—the potential beneficial effects of these compounds to palliate Se-induced stress were evaluated at physiological, biochemical and molecular levels. Se-treated plants accumulated Se in their roots (92%) and leaves (95%) more than control plants. Furthermore, Se diminished plant growth, photosynthetic traits and the relative water content and increased the levels of malondialdehyde, H_2_O_2_, osmolyte and endogenous NO. Exogenous Spm significantly decreased the levels of malondialdehyde by 28%, H_2_O_2_ by 37% and electrolyte leakage by 42%. Combined Spm/NO treatment reduced the Se content and triggered plant growth, photosynthetic traits, antioxidant enzymes and glyoxalase systems. Spm/NO also upregulated *MTP1*, *MTPC3* and *HSP70* and downregulated *TaPCS1* and *NRAMP1* (metal stress-related genes involved in selenium uptake, translocation and detoxification). However, the positive effects of Spm on Se-stressed plants were eliminated by the NO scavenger. Accordingly, data support the notion that Spm palliates selenium-induced oxidative stress since the induced NO elicits antioxidant defence upregulation but downregulates Se uptake and translocation. These findings pave the way for potential biotechnological approaches to supporting sustainable wheat crop production in selenium-contaminated areas.

## 1. Introduction

Wheat is the third most important cultivated cereal crop in the world [1] and it is also a great source of plant-based protein among the cereals [2]. Anthropogenically contaminated agricultural lands have excess metalloids, which present an emerging threat of great concern because of their overaccumulation in soil, long persistence in the environment and relative toxicity in all living organisms [3,4]. Selenium (Se) is a naturally occurring metalloid, and its phytotoxicity in plants depends on the dose, speciation and target species [5]. The major sources of Se are parent rocks, volcanic activities, agriculture practices, industries, etc. The two major forms of Se in soils are selenite and selenate, which are taken up by the roots of plants. Sodium selenate (Na_2_SeO_4_) is transported by phosphate transporters in plants and is phytotoxic at high concentrations [6]. In general, the Se content in soil ranges from 0.01–2 mg kg^−1^ [6]. Dai et al. [7] reported that soil Se toxicity occurs when the soil Se level exceeds 3.0 µg^−1^. The potential of plants to absorb Se in their tissues is proportional to whether they are non-accumulators (<0.1 g kg^−1^ DW), secondary accumulators (0.1–1.0 g kg^−1^ DW) or hyperaccumulators (>1.0 g kg^−1^ DW) [8].

Plants’ exposure to higher concentrations of selenium can trigger their overproduction of reactive oxygen species (ROS), which can induce oxidative damage and malfunction of essential micro- and macromolecules in plant cells [6,9,10,11]. To overcome oxidative stress injuries, plants have evolved strategic defence mechanisms, such as boosting the antioxidant system [4]. These defence mechanisms enhance the plant’s ability to neutralise the excess production of ROS to protect cells from oxidative stress. In response to oxidative stress, plants produce enzymatic antioxidants, such as peroxidase (POD), superoxide dismutase (SOD) isozymes, peroxiredoxins and catalase (CAT), as well as all components of the ascorbate-glutathione cycle including ascorbate peroxidase (APX), monodehydroascorbate reductase (MDHAR), dehydroascorbate reductase (DHAR) and glutathione reductase (GR) present in the main subcellular compartments [12]. Furthermore, plants have non-enzymatic antioxidants, including ascorbate (AsA), glutathione (GSH) and phenolic compounds, which participate in defence mechanisms under metalloid stress environments, including Se stress [13]. Plants also synthesise methylglyoxal (MG) under biotic and abiotic stresses, and excess formation of MG causes a reduction in lipids and proteins and damages the cell membrane [14].

When plants grow under stressful environments, they produce various kinds of phytohormones and growth regulators including polyamines to combat stress [15,16,17,18]. Polyamines are growth regulators with a low molecular weight that play critical roles in plant physiological and developmental functions and abiotic stress tolerance [11,19]. Polyamines consist of spermine (Spm), spermidine (Spd) and putrescine (Put), and their positive roles in plant physiological processes have been well documented by earlier studies [20]. Recently, the action of polyamines, including Spm, in the heavy metal stress response was briefly summarised in the literature [20]. Spm is involved in reducing different stressors, such as cadmium (Cd) and lead (Pb) stress in wheat [21], and copper (Cu) stress in sunflowers [22]. In this respect, Spm appears to be a potent antimetal stress biostimulator among numerous naturally occurring defence metabolites within plants [11]. Se stress-induced injury of plants can be alleviated by using biostimulators such as Spm, thereby improving Se stress tolerance in plants. Additionally, Spm can induce more nitric oxide (NO) biosynthesis than can other polyamines [23]. Nitric oxide (NO) is a free radical considered a defence-related signalling molecule that plays a vital role in plants under different abiotic stresses [24], including salt [12,25,26], heavy metal [7,27,28] and drought stresses [29,30]. However, the roles of Spm-induced endogenous NO in plants under heavy metal stress have not been well studied in the literature.

Metal tolerance proteins (MTPs), heat shock proteins (HSP70), phytochelatin synthase (TaPCS1) and natural resistance-associated macrophage protein (NRAMP1) are the key factors related to metal stress in plants. The MTP gene family encodes for proteins mainly located in the plasma membrane (PM), Golgi apparatus and endoplasmic reticulum and plays an important role in metal detoxification processes [31]. Heat shock proteins (HSP70) defend against cellular damage during oxidative stress and are involved in protein and DNA repair [32]. In *Arabidopsis thaliana*, transcriptional regulation of *HSP70* governing 24-epibrassinolide mediates arsenic uptake, translocation and tolerance [33]. Moreover, the overexpression of the *TaPCS1* genes may result in higher metal tolerance, as reported in several plants [34]. Nevertheless, Spm-mediated Se uptake, translocation and detoxification mediated by these genes have not been reported in the literature.

To our knowledge, the NO function in Spm-induced tolerance to Se toxicity is unknown. Therefore, this study has been designed to investigate/elucidate the physiological and biochemical mechanisms that contribute to the increase of Se tolerance in wheat by the exogenous application of Spm in combination with NO, with a particular emphasis on the functioning of the antioxidant and glyoxalase systems. In addition, using wheat as an agronomical plant model with great economical relevance, the present study examines the relationship among Spm, NO, ROS metabolism and metal stress-related genes under Se-induced toxicity. To accomplish our primary objective, our results provide evidence that Spm in combination with NO application protects Se-induced phytotoxicity in wheat seedlings, modulating antioxidant mechanisms and detoxifying MG.

## 2. Materials and Methods

### 2.1. Wheat Variety, Plant Growth Conditions and Treatments

The greenhouse experiment was conducted by using seeds of wheat (*Triticum aestivum* L. cv Sakha 95) obtained from the Agriculture Research Centre (ARC), Ministry of Agriculture, Egypt. Before planting, seeds were sterilised with 1% (*w*/*v*) sodium hypochlorite (NaOCl) solution, and then 10 seeds were sown in each plastic pot filled with 2 kg of a sterilised mixture of clay soil and compost (5:1, *v*:*v*) (Appendix A). Three uniform plants per pot were selected for further growth after germination. The plants were grown under a 12-h light period at 24 °C ± 2 and 65% humidity. The plants were watered once on alternative days and fertilised with Hoagland’s nutrient solution every week. Spm (1 mM) was applied to the foliage (30 mL) of the wheat seedlings for one week after germination, once on alternate days, before Se stress was applied. The dose of Spm was chosen from earlier studies, in which 1 mM Spm successfully alleviated copper (Cu) toxicity in wheat seedlings [22]. Afterwards, 1 mM sodium selenate (Na_2_SeO_4_) was applied with a nutrient solution for an additional five weeks. The Se dose was chosen from earlier studies initiating Se stress, in which the range of Na_2_SeO_4_ was 0.1–1.5 mM [13]. We chose 1 mM Na_2_SeO_4_ to initiate Se stress; the same solution was used in rice seedlings by Mostafa et al. [13]. Sodium nitroprusside (SNP, 0.1 mM) as the NO donor was applied separately or in combination with Spm once every other day. During the stress period, 0.1 mM 2-4-carboxyphenyl-4,4,5,5-tetramethylimidazoline-1-oxyl-3-oxide (cPTIO) was applied as the NO scavenger with the combination of Spm and SNP once on alternate days. Figure 1 shows the experimental design for the Se foliar treatment alone or in combination with other chemical compounds (Table 1) of the wheat plants.

The wheat seedlings were treated with Spm, cPTIO and SNP solutions dissolved in 0.01% tween-20, while non-stressed plants received only the 0.01% tween-20 solution (30 mL). The plants were harvested at 49 days for analysis of the physiological and biochemical parameters.

### 2.2. Selenium (Se) and Nitric Oxide (NO) Measurements

The Se content in wheat leaves and roots was measured by following the methods of Mostofa et al. [13]. An atomic absorption spectrophotometer (Z-5000; Hitachi, Japan) was used to quantify the Se content. The spectrophotometer was calibrated with a Se standard solution, and the Se contents were calculated by using Mostofa et al. [13] formula.

Nitric oxide (NO) was quantified according to the Griess reaction, which is based on the spontaneous oxidation of NO to nitrite under physiological conditions according to the method described by Kaya et al. [35]. A 0.6 g leaf sample was homogenised with 50 mM cold acetic acid (3 mL, pH 3.6) and zinc diacetate (4%). The leaf extract was centrifuged at 10,000× *g* for 15 min at 4 °C. The upper layer was collected from the extract and turned into a pellet that was washed with the extraction medium (1 mL) after centrifugation. Charcoal (0.1 g) was added to the supernatant and mixed well. The mixture was filtered and vortexed. Then, 1 mL of Griess reagent was added to the mixture, which was left for 30 min at room temperature. The absorbance was taken at 540 nm, and finally, the nitrite derived from NO was quantified.

### 2.3. Growth Measurements

Plant height (PH) was measured using a measuring tape after harvest. The fresh shoot and root weights were measured, and finally, both the shoot fresh weight (SFW) and root fresh weight (RFW) were calculated. The shoot dry weight (SDW) and root dry weight (RDW) were obtained after oven-drying at 80 °C for 48 h.

### 2.4. Photosynthetic Pigments and Relative Water Content (RWC)

The Chl content was determined spectrophotometrically by the method of Arnon [36]. The absorbance of the clear solvent was recorded at 663 and 645 nm using a UV/VIS spectrophotometer (Genway, Tokyo, Japan). The transpiration rate (E) and photosynthetic rate (Pn) were determined from the carbon dioxide consumption of plants using an infrared gas analyser system (TPS-2, USA). Water use efficiency (WUE) was calculated as the net CO_2_ assimilation rate (ACO_2_/E) and measured with a gas analyser system (TPS-2, Haverhill, MA, USA). The relative water content (RWC) in wheat leaves was determined by measuring the fresh weight (FW), dry weight (DW) and turgid weight (TW) following the recent method of Hasan et al. [4].

### 2.5. Malondialdehyde (MDA) and Hydrogen Peroxide (H_2_O_2_) Contents and Electrolyte Leakage (EL)

The malondialdehyde (MDA) content was determined using the thiobarbituric acid method according to Heath and Packer [37]. The MDA content was calculated based on its molar coefficient of absorbance of 155 mmol L^−1^cm^−1^ and expressed as nmol g^−1^ FW.

The hydrogen peroxide (H_2_O_2_) content was determined by the following methods of Kaya et al. [35]. Leaves (0.3 g) were homogenised in trichloroacetic acid (3 mL, 0.1% (*w*/*v*)), and the mixture was centrifuged at 12,000× *g* for 15 min at 4 °C. Then, 0.5 mL supernatant was collected and added to the reaction mixture of 0.1 M potassium phosphate buffer (pH 7.8, 0.5 mL) and 1 mL potassium iodide (1 M). The mixture was kept in the dark for 1 h. Finally, the mixture was ready to use for the determination of the H_2_O_2_ content, and the absorbance was measured at 390 nm. Electrolyte leakage (EL) was determined based on the methods of Kaya et al. [35].

### 2.6. Proline, Total Soluble Sugar (TSS) and Anthocyanin Contents

The proline (Pro) content was determined following the methods of Bates et al. [38]. A 0.5 g leaf sample was homogenised in sulfosalicylic acid (5 mL, 3%, *w*/*v*). Then, the sample was centrifuged for 20 min at 12,000× *g* at 4 °C. Equal volumes (2 mL) of supernatant, glacial acetic acid and acid-ninhydrin were mixed and heated at 100 °C for 1 h. Toluene (4 mL) was added to the mixture, and coloured chromophores were extracted. The Pro content was determined by the absorbance at 520 nm with reference to a standard solution.

The total soluble sugar (TSS) contents in wheat leaves were determined via a phenol-sulfuric acid assay followed by the method of Du Bois et al. [39]. Fresh leaf samples (0.5 g) were extracted by using 10 mL of 80% ethanol. The mixture was centrifuged, and the collected supernatant was added to 2.5 mL of phenol solution (5%, *v*/*v*) with sulfuric acid (0.5 mL). The mixture was transferred to a water bath for heat treatment and allowed to stand for 20 min. To end the reaction, the mixture was kept at room temperature. A standard curve was used to calculate the TSS content, and the absorbance was read at 490 nm. The anthocyanin content in leaves was determined according to the protocols of Mancinelli [40].

### 2.7. Antioxidant Enzyme Assays

Soluble proteins were determined by following the methods of Hasan et al. [4]. A 0.1 g wheat leaf sample was taken and homogenised in ice-cold extraction buffer containing 0.1 M phosphate buffer (pH 7.5), 0.5 mM EDTA and 1 mM PMSF. Then, the extraction was centrifuged at 16,000× *g* for 20 min at 4 °C. These derived samples were used to determine the protein content and enzyme activity. The protein content in leaf samples was measured according to the protocol originally developed by Bradford [41]. Bovine serum albumin (BSA) was used as a standard for protein quantification.

SOD activity was determined by the methods of Hasanuzzaman and Fujita [14]. The optical density (OD) was recorded at 560 nm, and the activity was expressed as U mg^−1^ protein. Catalase (CAT) activity was measured by noting the consumption of H_2_O_2_ for 30–90 sec of the reaction based on the methods of Aebi [42]. The absorbance was measured at 240 nm, and the activity was expressed as µmol min^−1^ mg^−1^ protein. For the determination of ascorbate peroxidase (APX) activity, the protocol of Nakano and Asada [43] was followed. The absorbance was measured at 290 nm, and APX activity was recorded as µmol min^−1^ mg^−1^ protein. Glutathione reductase (GR) activity was determined by the method of Carlberg and Mannervik [44] and recorded as U mg^−1^ protein. DHAR activity was determined by following the protocol of Nakano and Asada [43]. The OD was measured at 265 nm, and the activity was reported as U mg^−1^ protein. Monodehydroascorbate reductase (MDHAR) activity was analysed based on the methods of Hossain et al. [45]. The absorbance was recorded at 340 nm, and MDHAR activity was expressed as U mg^−1^ protein.

### 2.8. Ascorbate (AsA) and Glutathione (GSH) Levels

AsA levels were measured in a fresh leaf sample based on the methods of Hasan et al. [4]. Fresh leaf samples (0.5 g) were collected and extracted in 3 mL of a cold solution of 5% metaphosphoric acid and 1 M EDTA. The mixture was centrifuged at 11,500× *g* and the supernatant was pipetted out for AsA and GSH measurements. A total of 0.6 mL of 500 mM K-phosphate buffer pH 7.0 was used to neutralise 0.4 mL of aliquot, followed by 100 mM K-phosphate buffer pH 7.0 with a 0.5 unit of ascorbate oxidase. The reduced AsA absorbance measurements were taken at 265 nm.

The GSH content was quantified as suggested by Griffiths [46]. Accordingly, 0.6 mL of 500 mM K-phosphate buffer at pH 7.0 was used to neutralise a 0.4 mL aliquot. The absorption rate of NTB (2-nitro-5-thiobenzoic acid) created by the reduction of DTNB (5,5′-dithio-bis (2-nitrobenzoic acid) at 412 nm was used to estimate the GSH.

The GSH content was calculated by using the following equation: GSH = total GSH − GSSG, where GSSG corresponded to glutathione disulphide.

### 2.9. Glyoxalase Systems and Lactate Dehydrogenase (LDH) Activity

Glyoxalase I (Gly I) and Gly II activities were determined by using the protocols of Hossain et al. [47] and Hasan et al. [4]. For the measurements of Gly I activity, 100 mM potassium phosphate buffer (pH 7.0), 15 mM magnesium sulphate, 1.7 mM GSH and 3.5 mM MG were used in the reaction. The increase in absorbance was measured for 1 min at 240 nm, and the activity was calculated using the 3.37 mM^−1^ cm^−1^ extinction coefficient. For the activity of Gly II determinations, 100 mM Tris–HCl buffer (pH 7.2), 0.2 mM DTNB and 1 mM S-d-lactoylglutathione were used in the reaction. The Gly II activity was determined using the extinction coefficient of 13.6 mM^−1^ cm^−1^.

LDH (EC 1.1.1.27) activity was assayed based on the methods of Couldwell et al. [48] with slight modifications. Leaves (0.1 g) were collected and homogenised with ice in 500 µL LDH assay buffer. To remove the insoluble materials, the extraction was centrifuged at 10,000× *g* for 15 min at 4 °C. Then, 2–50 µL samples were added to duplicate wells of a well plate. The sample was brought to a final volume of 50 µL with LDH assay buffer. A nicotinamide adenine dinucleotide (NADH) standard curve was used to calculate the LDH activity. The absorbance was read at 340 nm and the activity was expressed as the nmol min^−1^ mg^−1^ protein.

### 2.10. Methylglyoxal (MG) and d-Lactate Contents

MG was quantified according to the method formerly developed by Hasanuzzaman and Fujita [14]. Then, 0.5 g leaves was taken and homogenised in 0.5 M perchloric acid (HCLO_4_) for 15 min. The extraction was centrifuged at 11,200× *g* for 10 min. Then, charcoal was added to the supernatants for decolourisation. The solution was centrifuged at 11,200× *g* for 10 min after neutralisation with saturated potassium carbonate. *N*-acetyl-l-cysteine (0.5 M) and K-P buffer (100 mM, pH 7.0) were added to the solution, which was incubated for 15 min. The absorbance was recorded at 288 nm. A standard graph was developed by using known concentrations of MG, and finally, the MG content was calculated. d-lactate was estimated spectrophotometrically by using bacterial d-lactate dehydrogenase (d-LDH) in a coupled reaction following the methods of Monošík et al. [49]. The reaction mixture (1 mL) consisted of 100 mM phosphate buffer (pH 7.5), 0.025 U diaphorase, 3-(4,5-dimethylthiazol-2-yl)-2,5-diphenyltetrazolium bromide (MTT) and 0.4 mM phosphorylating NAD^+^-dependent glyceraldehyde-3-phosphate dehydrogenase. The reaction was initiated by adding 0.25 U of d-LDH. The absorbance was noted at 565 nm. Stock D-lactate solution was used as a standard for the determination of d-lactate.

### 2.11. Gene Expression of Metal-Related Proteins

Total mRNA was isolated from 0.5 g wheat plant leaves from all treatment groups using a total RNA extraction kit (Sigma-Aldrich, St. Louis, MO, USA) based on the manufacturer’s method. The purified RNA was measured spectrophotometrically, and RNA reverse transcription was performed. The reaction mixture consisted of 2.5 μL 5× buffer, 2.5 μL 2.5 mM dNTPs, 2.5 μL MgCl_2_, 4 μL oligo (dT), oligo dT primer (10 pmL/μL), 2.5 μL RNA and 0.2 μL (5 unit/μL) reverse transcriptase (Promega, Gutenbergring 10, 69190 Walldorf, Germany). Primers for specific genes and housekeeping genes were used in real-time analysis with a Rotor-Gene 6000 (Qiagen, Hilden, Germany). The relative gene expression was measured by following the 2^−ΔΔCt^ methods of Livak and Schmittgen [50]. Specific gene accession numbers and the sequences of specific primers designed for qRT-PCR are listed in Appendix A.

### 2.12. Statistical Analysis

The TB tools statistic package was used to construct the heatmap. Principal component analysis (PCA) was performed in R version 3.6.3 using the ggbiplot package. All the obtained data were analysed by one-way analysis of variance (ANOVA) using Minitab 17.0 software (State College, PA, USA) and Fisher’s LSD test was conducted to test the significance between the mean values (*p* ≤ 0.05). Three biological replicates were performed for each treatment, and each replicate used at least three plants for each treatment to assess various parameters under the same experimental conditions.

## 3. Results

### 3.1. Se and NO Contents in Leaves and Roots of Wheat Seedlings

The Se content increased significantly in the leaves and roots of Se-stressed wheat seedlings by 95 and 92%, respectively, compared to those of the control plants. Nevertheless, the Se + Spm, Se + SNP, and Se + Spm + SNP treatments significantly decreased the Se contents in leaves and roots by 63, 58 and 68% and 53, 46 and 62%, respectively, compared to those in Se-stressed plants. The Se + Spm + SNP treatment was the most effective among the treatments in terms of reducing Se accumulation in wheat seedlings. In the wheat seedlings, the application of cPTIO overturned the positive effects of Spm, Spm + SNP or Spm + SNP on the leaf and root Se contents by increasing the Se contents (Figure 2A,B).

These results suggest that endogenous NO was required for the Se treatment to become an active participant in reducing Se accumulation in the leaves and roots of wheat seedlings. Se stress significantly decreased the NO content in leaves and roots by 45% and 68%, respectively, compared to that in the control plants (Figure 2C,D). Nevertheless, the Se + Spm, Se + SNP and Se + Spm + SNP treatments significantly increased the endogenous NO contents in leaves and roots by 82, 80 and 85% and 83, 82 and 86%, respectively, in comparison with those in the Se-stressed plants. Treatment with cPTIO along with Spm, SNP or Spm + SNP reduced the NO content in the leaves and roots of wheat seedlings, indicating that Spm might effectively participate in producing endogenous NO in wheat seedlings.

### 3.2. Spm and NO Alleviates Se Stress-Induced Growth Retardation

In comparison with the control, Se stress significantly decreased the PH, SFW, SDW, RFW and RDW by 44, 39, 52, 43 and 57%, respectively (Figure 3A–E).

Conversely, exogenous application of Spm markedly increased the PH by 27%, SFW by 21%, SDW by 34%, RFW by 26% and RDW by 47%. Moreover, the combination of Spm with NO (Se + SNP) increased these parameters by 26, 21, 33, 23 and 44%, respectively, compared with Se-stressed seedlings.

### 3.3. Spm-Induced Intrinsic NO Improves the RWC and Photosynthetic Pigments of Wheat Seedlings under Se Stress

Se stress significantly decreased the leaf RWC, total Chl, Pn, E and WUE in wheat plants by 28, 34, 18, 13 and 23%, respectively, in comparison with the control levels. On the other hand, the Se + Spm treatment led to an increase in the RWC by 17%, total Chl by 16%, Pn by 10%, E by 8% and WUE by 16% relative to Se stress (Figure 4A–E).

cPTIO negatively impacted those parameters related to the Spm and NO treatment combination, possibly by blocking NO. These results indicate that Spm induced NO synthesis to recover the relative water content and photosynthetic traits under Se toxicity.

### 3.4. Spm and NO Decreased the Content of MDA, H_2_O_2_ and EL of Wheat Seedlings under Se Stress

The contents of MDA, H_2_O_2_ and EL are displayed in Figure 5A–C. Se stress greatly increased the MDA, H_2_O_2_ and EL by 40, 56 and 58%, respectively, relative to the control levels. Nevertheless, pre-treatment with Se + Spm decreased the MDA content by 23%, the H_2_O_2_ content by 26% and EL by 38% in comparison with the Se-stressed plants. However, these parameters were significantly diminished by the application of Spm or SNP. The Se + SNP and Se + Spm + SNP treatments were also effective at decreasing these oxidative stress parameters by 23, 19 and 34%, and 29, 37 and 45%, respectively, relative to those in Se-stressed wheat seedlings (Figure 5A–C).

The positive effects of Spm, Spm + SNP and Se + Spm + SNP on these oxidative stress parameters were eliminated by the application of the NO scavenger cPTIO, showing that Spm-induced NO acts as a downstream signalling molecule that is involved in decreasing oxidative stress parameters under Se stress.

### 3.5. Spm and NO Decreased Proline and TSS Accumulation and Increased the Anthocyanin Content of Wheat Seedlings under Se Stress

Figure 5D–F shows the proline, total soluble sugar (TSS) and anthocyanin contents under Se stress. The Se treatment led to significant increases in proline and TSS by 45 and 42%, respectively, which in turn, noticeably decreased the anthocyanin content by 38% relative to the control. However, the Se + Spm, Se + SNP and Se + Spm + SNP treatments significantly reduced the proline and total soluble sugar (TSS) contents by 23, 19, and 24% and 19, 17, and 14%, respectively, and increased the anthocyanin content by 26, 25, and 24%, respectively, in comparison with the Se-stressed wheat seedlings. Furthermore, these tests showed that NO was essential for Spm-induced improvement of these parameters and that cPTIO reduced SA-, NO- and SA + SNP-related parameters.

### 3.6. Spm-Induced Endogenous NO Increased Antioxidant Enzyme Activities in Se-Stressed Wheat Seedlings

In the Se-stressed wheat plants, antioxidant enzyme activities such as those of SOD, CAT, APX, DHAR and MDHAR were increased by 25, 18, 8, 14 and 26%, respectively, in comparison to untreated plants (Figure 6A–F).

Surprisingly, the GR content declined under Se stress by 17% compared to the control level. Application of exogenous Spm and SNP restored the SOD, CAT, APX, GR, DHAR and MDHAR enzyme activities in Se-stressed wheat, resulting higher enzymatic activities in Se + Spm-treated seedlings by 19.1, 8, 7, 29, 19 and 20%, in Se + SNP-treated seedlings by 14, 4, 4, 23, 16, and 21% and in Se + Spm + SNP-treated seedlings by 22, 9, 8, 30, 27, and 20%, respectively, than seedlings subjected to Se stress alone. cPTIO treatment reversed the changes in SOD, CAT, APX, GR, DHAR and MDHAR activities and indicated that Spm-induced NO upregulated these enzymatic activities under stress conditions.

### 3.7. Spm and NO Improved the AsA-GSH Contents in Wheat Seedlings under Se Stress

Wheat plants under Se stress showed a 34% decrease in AsA and a 22% decrease in GSH compared to the control levels (Figure 6G–H). However, exposure of wheat plants to the Se + Spm, Se + SNP and Se + Spm + SNP treatments led to increases in AsA contents of 48, 43 and 50% and GSH contents of 30, 28, and 28%, respectively, compared with Se-stressed seedlings. Treatment with cPTIO along with Spm and SNP eliminated the increase in AsA and GSH contents. These results indicate that Spm-induced regulation of AsA and GSH was dependent on NO synthesis.

### 3.8. Spm and NO Decreased MG Intermediates and Improved Glyoxalase Systems

Exposure of wheat plants to Se stress caused a significant reduction in Gly II activities, by 33% compared with control seedlings, while the Gly I and LDH activities increased by 18 and 18%, respectively (Figure 7A–C).

In comparison with Se-stressed plants, Se + Spm, Se + SNP and Se + Spm + SNP-treated plants displayed significant increases in Gly I activity of 17, 11 and 22%, Gly II activity of 45, 44 and 50% and LDH activity of 19, 13 and 20%, respectively. The beneficial effects of Spm and the NO-donor SNP supplementation were eradicated by the NO scavenger (cPTIO), possibly due to high amounts of endogenous NO synthesis, which triggered upregulation of glyoxalase systems and lowered MG contents. Se stress triggered a significant increase in the MG and d-lactate contents in wheat seedlings by 50 and 24%, respectively, in relation to control plants (Figure 7D–E). The addition of Se + Spm, Se + SNP and Se + SPM + SNP resulted in a significant decline in MG content in wheat seedlings by 33, 25 and 40% and d-lactate by 10, 7, and 16%, respectively, compared to Se-stressed seedlings (Figure 7D).

### 3.9. Spm Induced NO-Modulated Gene Expression of Metal-Related Proteins under Se Stress

The relative transcript abundances of *MTP1*, *MTPC3* and *HSP70* were substantially increased in Se-stressed wheat plants, whereas the mRNA levels of *TaPCS1* and *NRAMP1* were significantly downregulated (Figure 8).

In addition, the combined application of Se + Spm, Se + SNP and Se + SPM + SNP further led to enhancement of the relative transcript abundance of *MTP1* by 43, 46 and 42%, *MTPC3* by 42, 44 and 43% and *HSP70* by 44, 45 and 43%, respectively, and decreased the *TaPCS1* and *NRAMP1* transcript levels by 35, 33, and 34% and 45, 47, and 47%, respectively, relative to those of Se-stressed wheat seedlings. However, cPTIO treatments markedly upregulated *MTP1*, *MTPC3* and *HSP70* and downregulated *TaPCS1* and *NRAMP1* gene expression in wheat seedlings compared to Se-stressed seedlings. These results imply that the effects of Spm and SNP supplementation might be inactivated by the blockade of Spm-induced NO caused by cPTIO application.

### 3.10. Heat Map Analysis and Principal Component Analysis (PCA) of Spm- and NO-Treated Wheat Seedlings under Se Stress

The mean values of physiological and biochemical parameters were applied to create a heatmap. The log 10 values of these parameters clearly revealed differences between traits represented by colour intensities ranging from low to high (Figure 9A,B).

Moreover, the heatmap shows Se-stressed plants separated from control seedlings and Spm-, SNP- and cPTIO-treated seedlings. PCA was performed to determine the growth and physiological parameters of the eight treatment groups (Control, Se, Se + Spm, Se + SNP, Se + Spm + SNP, Se + Spm + cPTIO, Se + SNP + cPTIO, Se + Spm + SNP + cPTIO). The treatment separations with their biological replicates were clearly identified by analysing the PCA biplot (Figure 10). In the dataset, PC1 and PC2 had the highest contributions and together accounted for 99.42% of the total variance, with PC2 contributing 2.41%, while PC1 contributed 97.01% alone.

## 4. Discussion

### 4.1. Spm-Induced Endogenous NO Diminishes the Se Content in the Leaves and Roots of Se-Stressed Wheat

Endogenous NO was analysed to evaluate whether this molecule participates in physiological enhancements in wheat plants under Se toxicity. The data showed that the endogenous NO content declined under Se stress; however, Spm promoted NO accumulation under Se stress. A comparable study was reported in maize seedlings under arsenic stress, where salicylic acid (SA) mediated NO regulation [35]. Consequently, the data suggest that NO acts as a key signalling molecule during Se toxicity. Endogenous NO was also found to increase when plants were exogenously supplemented with Spm and NO-donor SNP, which supports the notion that NO might modulate ROS generation by upregulating the antioxidant and glyoxalase systems in wheat. In contrast, the balance of endogenous NO production is an important issue in the case of stress tolerance. A certain level of endogenous NO helps to improve abiotic stress tolerance, whereas excess generation of NO can hamper plant growth and development [35]. In our experimental model, the data show that the NO induced by Spm did not exceed the toxic levels in wheat plants under Se stress. The data also suggest that the Spm is more efficient at generating NO in wheat than the NO-donor SNP.

On the other hand, Se-treated wheat plants had higher Se contents in both their leaves and roots, as has also been reported in the literature in roots, leaf sheaths and blades of rice plants [13]. Likewise, it has been reported that Spm mediated a reduction of heavy metal accumulation in sunflower and wheat plants [21,51]. Kaya et al. [35] reported that H_2_O_2_ improves metal ion influx, whereas antioxidant enzymes decrease metal influx. Thus, an excess of H_2_O_2_ seems to be involved in plasma membrane leakage through oxidative damage of lipids and proteins. As a result, a reduction in APX was observed under Se toxicity, and more Se easily accessed the cell. Nonetheless, the exogenous application of Spm or Spm + SNP greatly reduced Se accumulation in roots and leaves, which indicates that Spm decreases excess Se uptake and translocation. This could be due to restoration of the antioxidant activities of wheat seedlings and Spm-modulated downregulation of root-to-shoot Se transporters. Taie et al. [21] reported similar results, showing that Spm restricts the access of metal ions (Cd^2+^ and Pb^2+^) to wheat plants. Hence, Spm itself or Spm + SNP mitigates Se toxicity in wheat plants where NO appears to be of relevance in terms of participation. This is supported by how the NO scavenger cPTIO treatment overturned the decrease in Se content. It was also observed that the suppression of Se in both roots and leaves was due to Spm-induced NO synthesis. Analogous observations were found in rice plants, where exogenous NO donor SNP diminished the arsenic content in leaves [13].

### 4.2. Spm-Induced NO Improved Growth, and Photosynthetic Performances under Se Stress

Se also triggered detrimental effects on wheat growth parameters, which is in good agreement with previous studies in other crops, including *Brassica napus* [6] and *Oryza sativa* [7]. However, Spm treatment palliated these harmful effects of selenium in wheat seedlings. Photosynthesis is usually affected by multiples biotic stresses [52], and Se stress suppressed the photosynthesis rate of wheat seedlings as in *Brassica napus* [6]. However, Spm pre-treatment reduced this damage to the functioning of photosynthesis of wheat plants, as has been described previously with cadmium and lead [21]. This effect may be due to the decrease in H_2_O_2_ content when Spm was applied, as previously reported in sunflowers [26]. NO positively regulated photosynthesis and increased the chlorophyll content, as described in tomato plants [53]. Similar to previous studies, our data showed the enhancement of photosynthetic traits after NO-donor SNP treatment along with Se-induced stress.

### 4.3. Spm-Induced NO Decreases Oxidative Damage and Improves Antioxidant Defense under Se Stress

Uncontrolled ROS generation can cause oxidative damage to cellular macromolecules, resulting in membrane deterioration and cell death [54]. Greater accumulations of H_2_O_2_ and MDA were observed in wheat leaf tissues under Se stress, which were associated with lipid peroxidation and membrane damage. Similar findings have been reported in rice under severe Se stress [7]. Nevertheless, Spm and Spm + SNP pre-treatments strikingly decreased the H_2_O_2_, MDA and EL levels, suggesting that Spm alleviates Se-induced oxidative stress in wheat seedlings. The NO implication of mitigating oxidative damage has also been reported in rice and maize under other heavy metal stresses [35,55]. Se toxicity triggered a reduction of the RWC in wheat seedlings, as has been reported in rice plants [13]. However, treatment with Spm and SNP positively restored the RWC content in wheat, while the NO scavenger cPTIO reversed this effect. Proline is an important osmolyte that regulates abiotic stress tolerance in plants [4], and under Se stress, wheat plants had a higher level of proline accumulation, demonstrating a similar response to that found in rice seedlings [7]. Anthocyanins are coloured water-soluble pigments that are synthesised through the phenylpropanoid pathway; they have antioxidant properties because anthocyanins can scavenge ROS such as superoxide anions (O_2_^•−^) and H_2_O_2_ [56]. The addition of Spm triggered an increase in the anthocyanin content and consequently an increase of the antioxidant capacity, and accordingly, the content of H_2_O_2_ was found to be lower. However, cPTIO treatment of wheat seedlings suppressed the endogenous NO content, indicating that anthocyanin production depends on Spm and SNP under Se toxicity.

Under oxidative stress, plants activate diverse antioxidant systems to combat the adverse effects of uncontrolled excess ROS production [11]. In the present study, Spm upregulated the analysed antioxidant enzymes, which lowered the ROS content. SOD is considered the first line of the defence of the antioxidant enzymatic system [12,57], and SOD activity increased in wheat seedlings under Se-induced stress. Furthermore, Spm treatment intensified this effect on SOD, in a similar fashion to the process observed in wheat plants under other metal stresses [21]. In a previous study, exogenous NO diminished salinity-induced oxidative stress in tomato plants with a concomitant increase of total SOD activity in leaves and roots. A deeper analysis of the different SOD isozymes showed that the MnSOD activity was unaffected, whereas the activity of two CuZnSOD isozymes increased in both organs, with the noteworthy induction of a new CuZnSOD (III) in roots [58].

Catalase (CAT) activity increased in wheat plants under Se stress, as has been described in the cowpea [59]. During Se-induced stress, plants supplied with Spm and SNP had higher CAT activity. These findings are in line with others for rice, as described by Mostafa et al. [13]. APX activity also increased in the wheat plants under Se stress, as previously reported in rice seedlings [13]. Overaccumulation of H_2_O_2_ in different subcellular compartments can lead to protein and lipid damage due to inhibition of APX activity. The Spm and Spm + SNP treatments that triggered the increase of APX activity in the wheat seedlings seem to be mediated by NO. This is supported by previous reports that have proven that APX activity can increase by a process of S-nitrosation [35]; as well as this, APX activity increased in pepper fruits exposed to an enriched NO atmosphere [60].

Se stress reduced the GR activity in wheat seedlings, as previously described in maize under As stress [35]. Interestingly, the GR activity was found to be upregulated after the application of Spm and Spm + SNP treatments in wheat plants, possibly due to NADPH availability, which is essential for GR activity. This is in good agreement with previous studies where the NADPH-generating enzymatic system has been shown to be essential for maintaining cellular redox stability under stress [12]. Similarly, Se stress increased the DHAR and MDHAR activity in wheat plants in the current study. However, an increase in DHAR and MDHAR activity was recorded in the present study in wheat plants under the Spm and Spm + SNP treatments. Ahmad et al. [53] reported that NO upregulated DHAR activities in *Solanum lycopersicum* under Cd stress, indicating that Spm induced a NO role in antioxidant systems. Similar observations were found in rice seedlings under Cu stress [56].

As a result, Spm and Spm + SNP treatments increased the activities of antioxidant enzymes, including SOD, CAT, APX, GR, DHAR and MDHAR, which are present in the main subcellular compartments, thus avoiding ROS generation in these organelles [61,62,63], and consequently, diminishing the oxidative damage throughout the cell and maintaining its viability. However, NO scavenger cPTIO reversed the antioxidant enzyme activities, which indicates that cPTIO might inhibit endogenous NO accumulation in the leaves. These results suggest that NO directly contributes to Spm-mediated antioxidant enzyme upregulation to increase the Se tolerance in wheat. Subsequently, Spm may scavenge excessive ROS by triggering NO, and may improve plant growth performance by modulating antioxidant systems.

AsA and GSH are the most abundant soluble non-enzymatic antioxidants that maintain the cellular redox status [64]. Se induced a decline in AsA content in wheat such as has been reported in rice [13]. In addition, several studies reported a decrease in the contents of AsA and GSH in many crops under metalloid stress, e.g., arsenic stress in maize [35] and copper stress in rice [56]. Strikingly, both the Spm and Spm + SNP treatments restored the redox status of AsA and GSH pools, possibly by improving their redox status and cell membrane protection. However, NO scavenger cPTIO blocked the NO-induced by Spm, resulting in inhibition of Spm-induced AsA and GSH upregulation. These results suggest that NO induced by Spm is required for the activation of AsA-GSH pools to improve Se stress tolerance.

Methylglyoxal (MG) systems depend on the enzymes Gly I, Gly II and LDH under abiotic stress [14]. These enzymes are required to minimize the excess generation of MG, resulting in improved abiotic stress tolerance. MG and d-lactate are two key intermediates involved in MG systems. Our data suggest that overaccumulation of Se in the roots and leaves of wheat caused excessive MG production, possibly due to a significant reduction in Gly II activity. Similarly high MG accumulation was observed in rice under Se toxicity [13]. However, Spm-induced endogenous NO significantly increased Gly I, Gly II and LDH enzyme activities and decreased MG and d-lactate contents, indicating effective MG detoxification and enhanced tolerance of Se stress. However, cPTIO inhibited the glyoxalase systems and MG detoxification by blocking NO accumulation/synthesis in wheat seedlings under Se stress. Our results suggest that Spm-induced upregulation of glyoxalase system enzymes and MG detoxification relies on NO synthesis. Hasanuzzaman and Fujita [14] reported comparable effects, where the exogenous application of NO improved the glyoxalase systems in wheat.

### 4.4. Modulation of Gene Expression in Spm and NO-Treated Wheat Seedlings under Se Stress

Metal tolerance proteins (MTPs) are a class of membrane proteins that act as efflux transporters from the cytoplasm of the cell. In wheat, the expression of *MTP 1* and *MTP C3* was remarkably upregulated in Se-stressed plants. These findings are consistent with previous experimental results in which the transcript levels of *MTP 1* increased in the shoots of rice seedlings under cobalt (Co), cadmium (Cd) and nickel (Ni) stress [65]. Moreover, exogenous application of Spm further elevated both of these metal tolerance proteins in Se-treated seedlings. Salicylic acid-mediated upregulation of the *MTP* gene family was found under Zn stress in sunflower seeds [66], which also supports our findings. Consequently, Spm-induced NO seems to regulate the expression of *MTP 1* and *MTP C3*, and thus, participate in the discharge and vacuole segregation process of Se.

Heat shock protein (HSP) adaptation is an adaptive response to heavy metal stresses in plants. This study showed that *HSP70* gene expression was upregulated in wheat seedlings under Se stress, whereas Spm significantly improved the transcript level of *HSP70.* Luo et al. [67] stated that polyamines such as spermidine (Spd) led to increased transcript levels of the *HSP70* gene under heat stress, and this finding is in good agreement with our data. These results suggest that Spm-induced NO is involved in *HSP70* expression, which may function in HSP accumulation, stabilisation of denatured proteins and accurate protein-folding in wheat seedlings under Se-induced oxidative stress.

Selenium stress upregulated *TaPCS1* expression, which encodes for phytochelatin synthase (PCS). In contrast, Spm treatment led to downregulation of the transcript levels of *NRAMP1*, suggesting that a strong interaction with Spm induced NO to mediate Se uptake and transport. In our study, *NRAMP1* gene expression was upregulated in wheat seedlings under Se stress. However, Spm treatment downregulated *NRAMP1* transcript levels, which suggests that the inhibition of this gene expression led to decreased Se accumulation in wheat leaves and the roots of wheat seedlings. Previous studies support these findings; for instance, exogenous melatonin decreased the mRNA levels of *NRAMP1* in *M. baccata* under Cd stress [68]. However, cPTIO treatment triggered upregulation of the *MTP1, MTPC,* and *HSP70* genes and downregulation of the *TaPCS1* and *NRAMP1* genes. Thus, cPTIO treatment blocked Spm-induced endogenous NO synthesis, resulting in the upregulation of *MTP1, MTPC3* and *HSP70* and the downregulation of *TaPCS1* and *NRAMP1*.

As presented above, the proposed mechanisms show that NO acts as a downstream signalling molecule of Spm to achieve Spm-induced Se tolerance in wheat plants by improving enzymatic and non-enzymatic components of the AsA-GSH cycle and glyoxalase system intermediates (Figure 10).

The model also shows that cPTIO counters Se-induced stress tolerance by decreasing NO and downregulating the antioxidant and glyoxalase systems. The current study’s extensive findings lead us to infer that Se is a phytotoxic agent at high concentrations and that Spm may be beneficial for mitigating excessive unfavourable Se-induced effects in wheat via modulation of numerous physio-biochemical and molecular processes. Finally, we have proven that exogenous application of Spm in combination with NO helps wheat plants to cope with selenium-induced oxidative stress by concurrently stimulating antioxidant and glyoxalase system components.

## 5. Conclusions

The current study concludes that excessive Se is a phytotoxic agent and proposes that Spm could represent an effective chemical for mitigating the detrimental impacts of increased Se in wheat by influencing many physiological, biochemical and molecular processes. Our data indicate that exogenous application of Spm in combination with NO increases Se stress tolerance and improves the growth and photosynthetic traits. Spm and NO donors play a key role in the alleviation of Se toxicity in wheat plants by modulating antioxidant activities, glyoxalase systems and MG intermediates. This study clearly demonstrates that applying Spm to wheat plants exposed to high levels of Se efficiently helps the plants to combat Se-phytotoxicity. Considering the economic relevance worldwide of wheat crops, the present data indicate that Spm and NO could act as potential biotechnological tools to provide routes to sustainable wheat crop production in Se-contaminated areas. Moreover, this research also provides basic biochemical knowledge that will help to decipher the mechanisms of crosstalk between Spm and NO at the physiological and molecular levels.

## Figures and Tables

**Figure 1 antioxidants-10-01835-f001:**
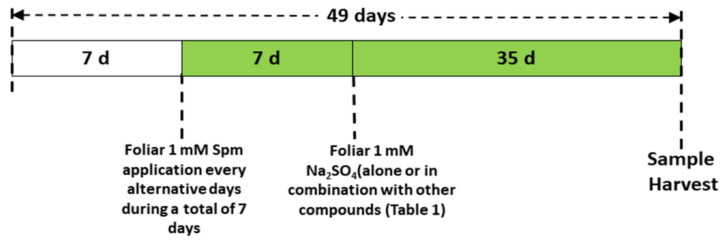
Scheme of the foliar treatments used to study the effects on wheat plants grown in the presence of 1 mM Na_2_SeO_4_.

**Figure 2 antioxidants-10-01835-f002:**
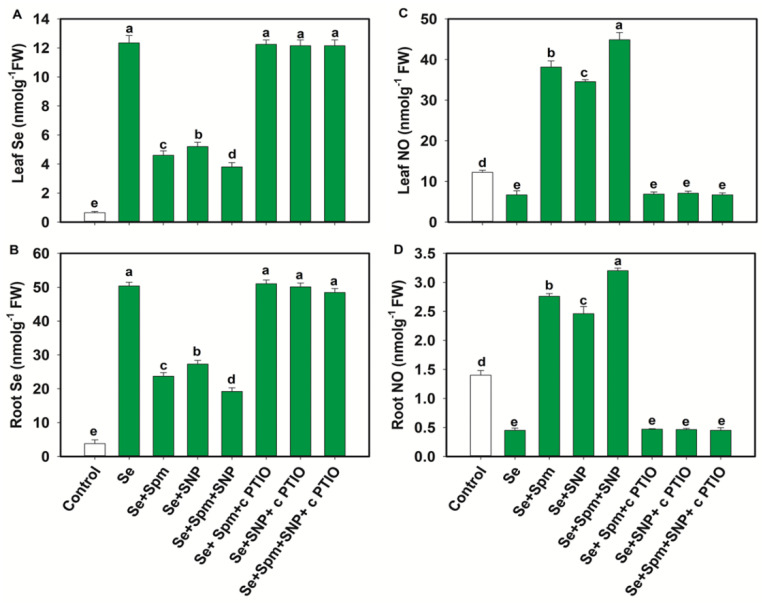
Selenium (Se) and NO contents (**A**–**D**) of leaves and roots of 49-day-old wheat plants grown under control and Se stress (1 mM) sprayed with 1.0 mM Spm or 0.1 mM SNP alone or together, combined with 0.1 mM cPTIO, a NO scavenger. Data presented are the mean (±SE) of three replicates, and bars with dissimilar letters (a–e) represent significantly different results at the *p* ≤ 0.05 level based on Fisher’s LSD test.

**Figure 3 antioxidants-10-01835-f003:**
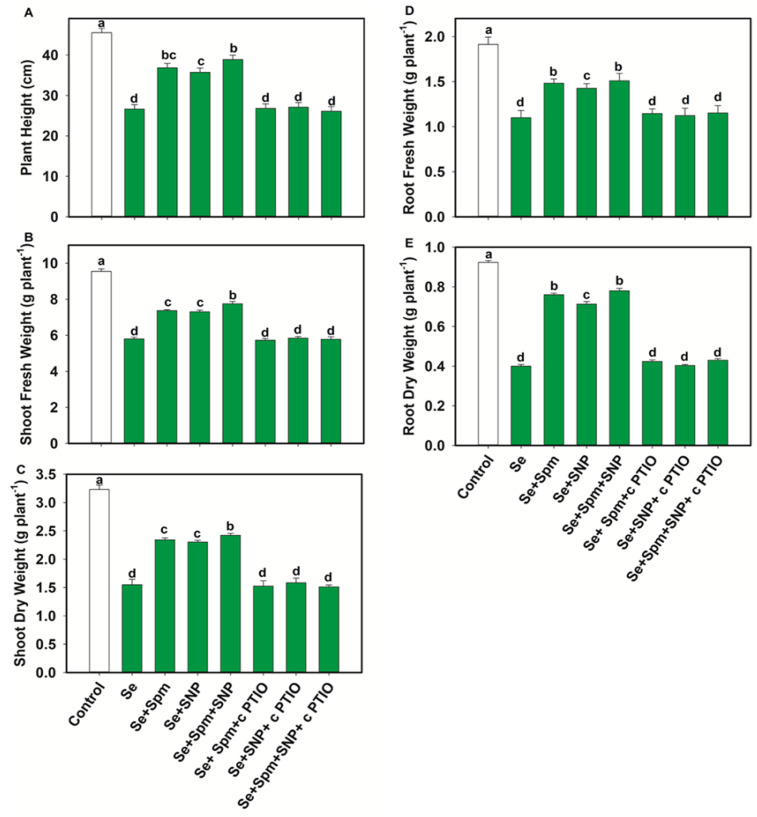
Plant height (**A**), shoot fresh weight (**B**), shoot dry weight (**C**), root fresh weight (**D**) and root dry weight (**E**) in leaves of 49-days-old-wheat plants grown under control and Se stress (1 mM) sprayed with 1.0 mM Spm or 0.1 mM SNP alone or together, combined with 0.1 mM cPTIO, a NO scavenger. Data presented are the mean (±SE) of three replicates, and bars with dissimilar letters (a–d) represent significantly different results at the *p* ≤ 0.05 level based on Fisher’s LSD test.

**Figure 4 antioxidants-10-01835-f004:**
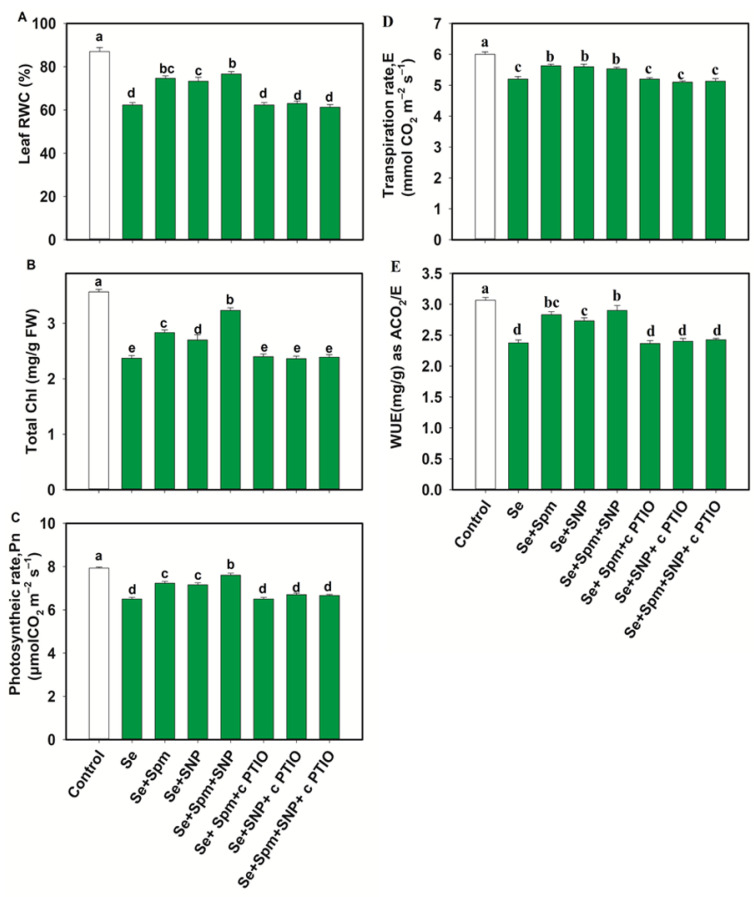
Leaf relative water content (RWC) (**A**), total chlorophyll (Chl) (**B**), photosynthetic rate (Pn) (**C**), transpiration (**D**,**E**) and water use efficiency (WUE) (**E**) in leaves of 49-day-old wheat plants grown under control and Se stress (1 mM) sprayed with 1.0 mM Spm or 0.1 mM SNP alone or together, combined with 0.1 mM cPTIO, a NO scavenger. Data presented are the mean (±SE) of three replicates, and bars with dissimilar letters (a–d) represent significantly different results at the *p* ≤ 0.05 level based on Fisher’s LSD test.

**Figure 5 antioxidants-10-01835-f005:**
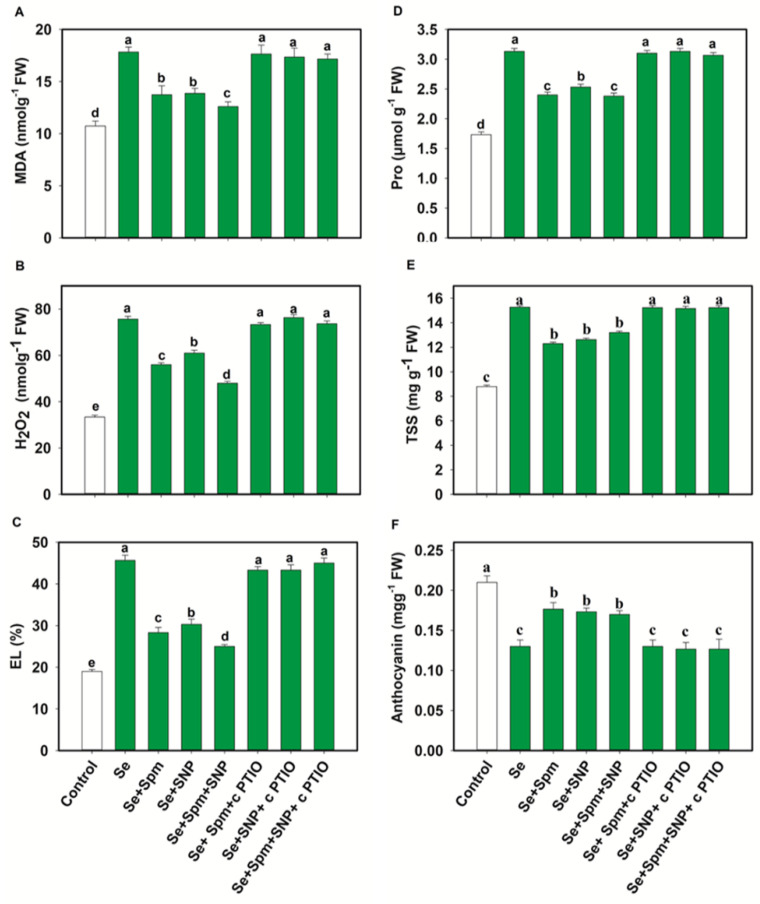
Malondialdehyde content (MDA) (**A**), hydrogen peroxide content (H_2_O_2_) (**B**). electrolyte leakage (EL) (**C**), Proline (Pro) (**D**), total soluble sugar (TSS) (**E**) and anthocyanin contents (**F**) of leaves of 49-day-old wheat plants grown under control and Se stress (1 mM) sprayed with 1.0 mM Spm or 0.1 mM SNP alone or together, combined with 0.1 mM cPTIO, a NO scavenger. Data presented are the mean (±SE) of three replicates, and bars with dissimilar letters (a–d) represent significantly different results at the *p* ≤ 0.05 level based on Fisher’s LSD test.

**Figure 6 antioxidants-10-01835-f006:**
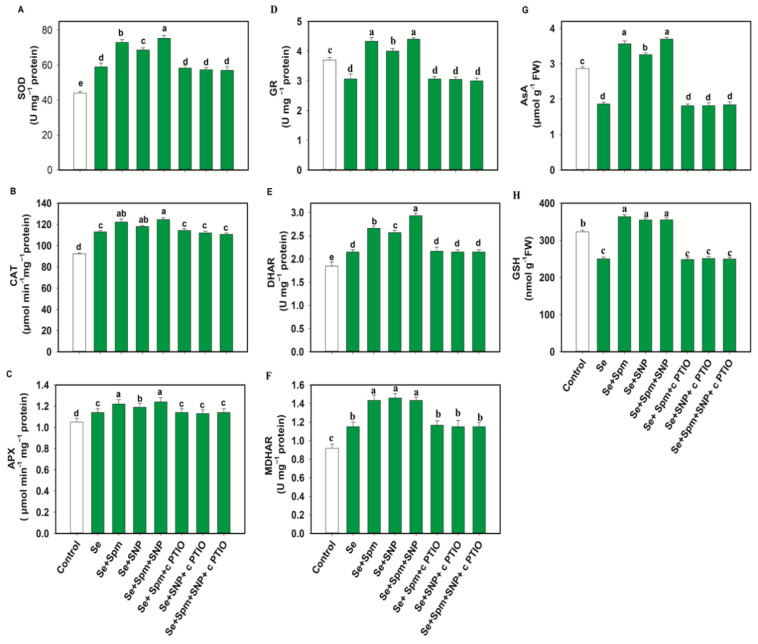
Superoxide dismutase (SOD) (**A**), catalase (CAT) (**B**), ascorbate peroxidase (APX) (**C**), glutathione reductase (GR) (**D**) dehydroascorbate reductase (DHAR) (**E**) and monodehydroascorbate reductase (MDHAR) (**F**) activities and ascorbate (AsA) (**G**) and glutathione (GSH) (**H**) levels in leaves of 49-day-old wheat plants grown under control and Se stress (1 mM) sprayed with 1.0 mM Spm or 0.1 mM SNP alone or together, combined with 0.1 mM cPTIO, a NO scavenger. Data presented are the mean (±SE) of three replicates, and bars with dissimilar letters (a–e) represent significantly different results at the *p* ≤ 0.05 level based on Fisher’s LSD test.

**Figure 7 antioxidants-10-01835-f007:**
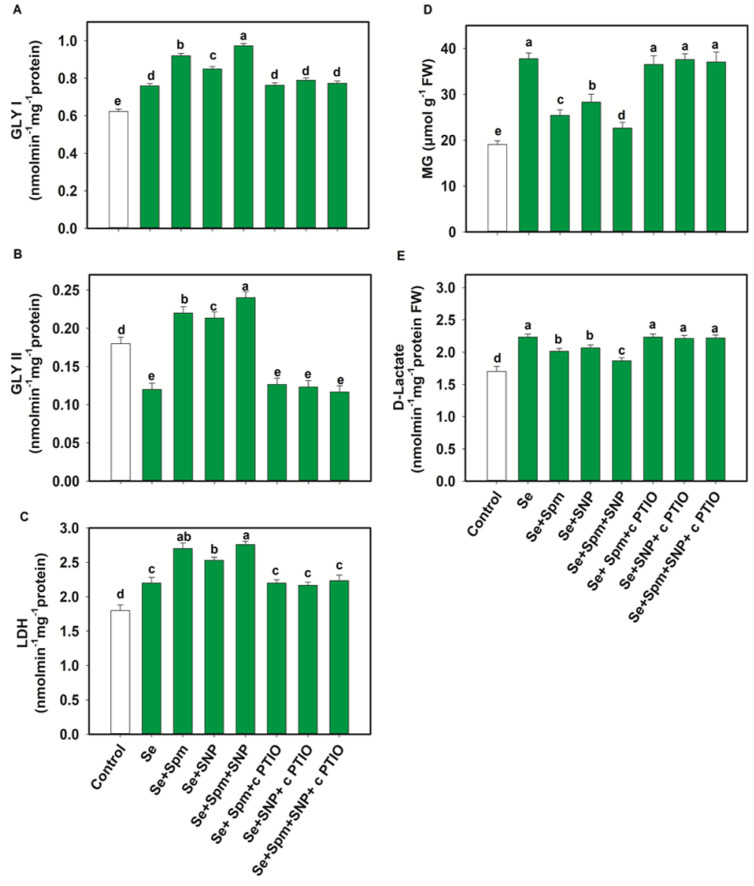
Glyoxalase I (GLY I) (**A**), glyoxalase II (GLY II) (**B**) and lactate dehydrogenase (LDH) activities (**C**) and methylglyoxal (MG) (**D**) and d-lactate (**E**) levels in leaves of 49-day-old wheat plants grown under control and Se stress (1 mM) sprayed with 1.0 mM Spm or 0.1 mM SNP alone or together, combined with 0.1 mM cPTIO, a NO scavenger. Data presented are the mean (±SE) of three replicates, and bars with dissimilar letters (a–e) represent significantly different results at the *p* ≤ 0.05 level based on Fisher’s LSD test.

**Figure 8 antioxidants-10-01835-f008:**
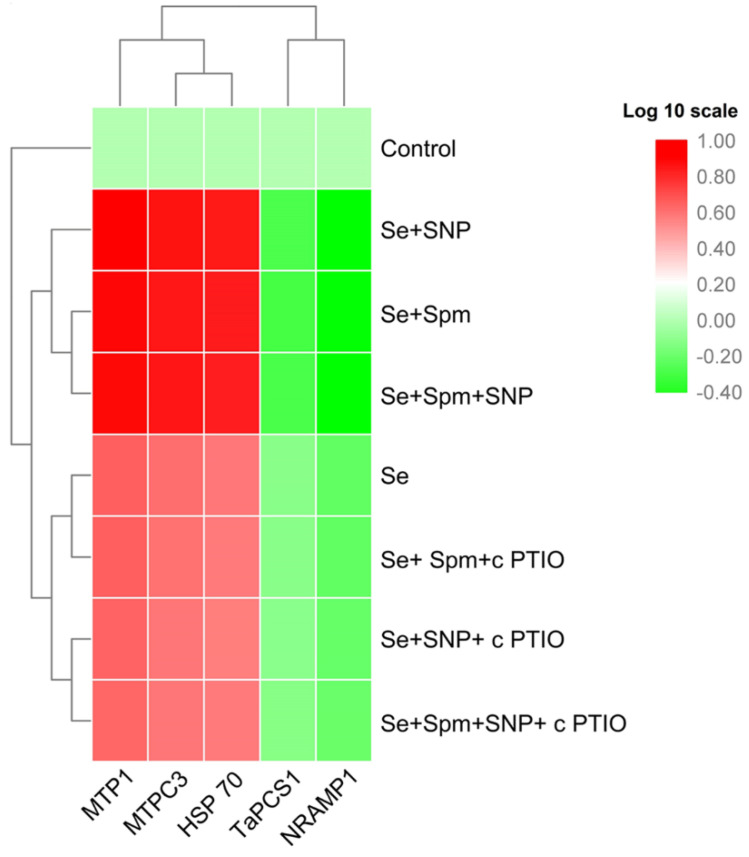
Heat map representing the relative transcript abundance of differentially expressed heavy metal-related genes in leaves of 49-day-old wheat plants grown under control and Se stress (1 mM) sprayed with 1.0 mM Spm or 0.1 mM SNP alone or together, combined with 0.1 mM cPTIO, a NO scavenger, and hierarchical cluster analysis results. The gene expression intensity is represented by colours from green (low) to red (high).

**Figure 9 antioxidants-10-01835-f009:**
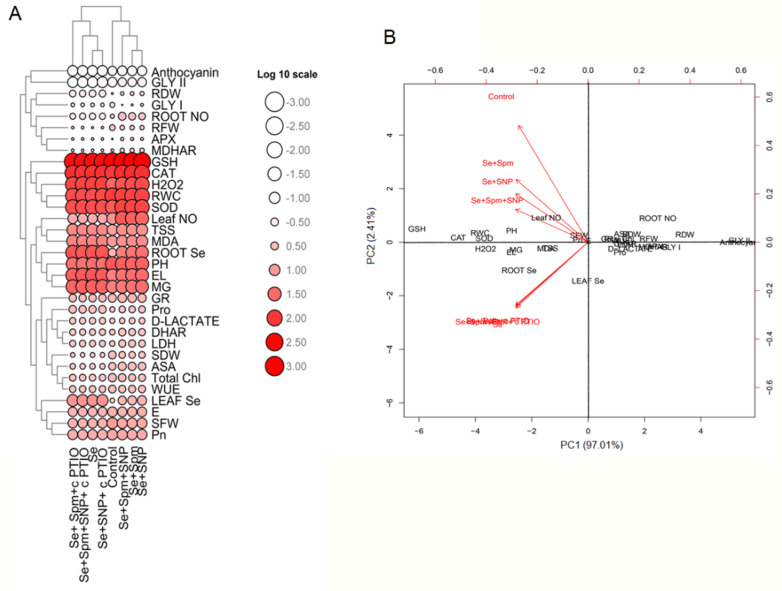
Heatmap (**A**) and PCA (**B**) displaying the variable-treatment relationships. In the heatmap, logarithmic (log10) values are shown on a colour scale (lower to higher values are represented by white to red). All datasets were subjected to PCA. The variables included leaf Se, leaf NO, root Se, root NO, PH, SFW, SDW, RFW, RDW, RWC, total Chl, Pn, E, WUE, MDA, H_2_O_2_ content, EL, Pro content, TSS content, anthocyanin, activity of SOD, CAT, APX, DHAR, MDHAR, AsA content, GSH content, MG, d-lactate, activity of Gly I, Gly II and LDH.

**Figure 10 antioxidants-10-01835-f010:**
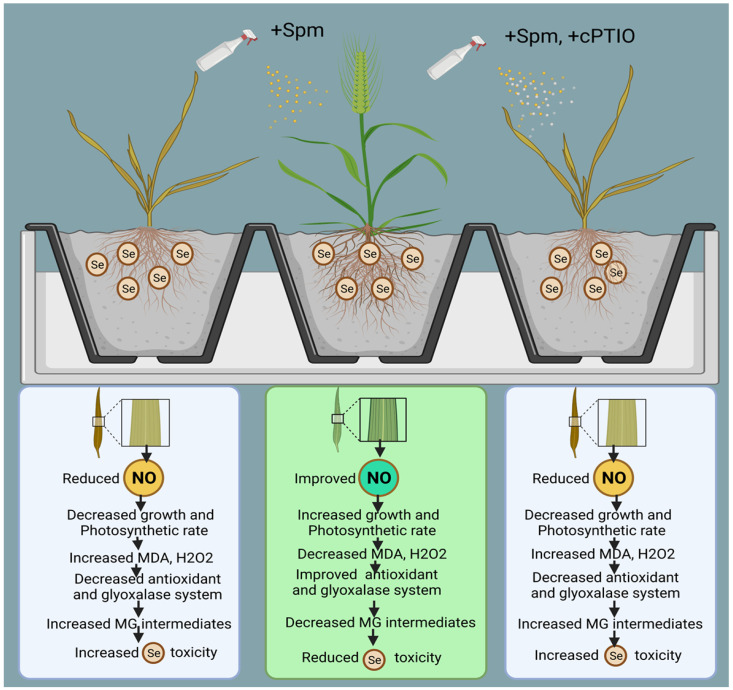
Model showing Spm-induced NO regulation in Se-stressed wheat plant.

**Table 1 antioxidants-10-01835-t001:** Foliar treatments used in this study.

Treatment	Na_2_SeO_4_ (mM)	Spm (mM)	SNP (mM)	cPTIO (mM)
Control	0.0	0.0	0.0	0.0
Se	1.0	0.0	0.0	0.0
Se + Spm	1.0	1.0	0.0	0.0
Se + SNP	1.0	0.0	0.1	0.0
Se + Spm + SNP	1.0	1.0	0.1	0.0
Se + Spm + cPTIO	1.0	1.0	0.0	0.1
Se + SNP + cPTIO	1.0	0.0	0.1	0.1
Se + Spm + SNP + cPTIO	1.0	1.0	0.1	0.1

## Data Availability

Not applicable.

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
