# Peer review of "Spermine-Mediated Tolerance to Selenium Toxicity in Wheat (Triticum aestivum L.) Depends on Endogenous Nitric Oxide Synthesis"

_antioxidants, 2021, doi:10.3390/antiox10111835_

Round 1
Reviewer 1 Report
In this study wheat plants were treated by 1 mM Se and its accumulation and effects were reduced by application of different polyamines (PAs) alone or in combination. The results indicated the role of NO in responses to Se and, besides the positive effect of PA spermine (Spm) and NO donor SNP, the quenching effect of the NO scavengers also proved this interesting finding. Several growth parameters, physiological process and the expression of selected stress marker genes were investigated. Principal Component Analysis of the dataset and a heat map illustrate the different mechanisms and the astonishing results.
Questions and critical remarks:
- In Abstract (lines 33-34) it is written: „ Se-treated plants accumulate Se in roots(92%) and leaves (95%)compared to control plants.” In Results (lines 300 -301) it is stated: „The Se content increased significantly in the leaves and roots of Se-stressed wheat seedlings by 95% and 92%, respectively, compared to those of the control plants” - According to Fig. 1. it is a much higher accumulation - check the calculation.
- In Figure 9 it should be sign that Spm and SPM+cPTIO added exogenously. I suggest supplementing with “+” or with other sign, for example indicating the spraying
- line 60-61: „ Dai et al. (2020) [7] reported that soil Se toxicity occurs when the soil Se level exceeds 3.0 µg-1” – Correct the unit! Check the format of reference!
- The manuscript needs a careful perusal. Check the format of reference through the manuscript. Most frequently the year should be deleted, but somewhere the numbering is missing.
Some other examples to correct:
- line 72: include
- lines 135, 136: Mostafa et al., 2020. On other sites (lines 151, 153) Mostofa et al. 2020 is written. Check and correct it!
- Uniform the writing format of µL and mL. In some cases (e.g., in sections 2.8 and 2.11) „µl” is used.
- line 552: (SA) mediated
- line 565: SPM mediated
- line 598: associated with
Author Response
In this study wheat plants were treated by 1 mM Se and its accumulation and effects were reduced by application of different polyamines (PAs) alone or in combination. The results indicated the role of NO in responses to Se and, besides the positive effect of PA spermine (Spm) and NO donor SNP, the quenching effect of the NO scavengers also proved this interesting finding. Several growth parameters, physiological process and the expression of selected stress marker genes were investigated. Principal Component Analysis of the dataset and a heat map illustrate the different mechanisms and the astonishing results.
Response: Thank you for the overall positive assessments of our manuscript.
Questions and critical remarks:
- In Abstract (lines 33-34) it is written: „ Se-treated plants accumulate Se in roots (92%) and leaves (95%) compared to control plants.” In Results (lines 300 -301) it is stated: „The Se content increased significantly in the leaves and roots of Se-stressed wheat seedlings by 95% and 92%, respectively, compared to those of the control plants” - According to Fig. 1. it is a much higher accumulation - check the calculation.
Response: We have fixed this error in the percentage (leaves-95%, roots-92%). Please check now.
- In Figure 9 it should be sign that Spm and SPM+cPTIO added exogenously. I suggest supplementing with “+” or with other sign, for example indicating the spraying.
Response: Now this figure is the number 10 and we have added the “+” sign. In addition, according to Reviewer 2, we have revised and added new details in the figure.
- line 60-61: „ Dai et al. (2020) [7] reported that soil Se toxicity occurs when the soil Se level exceeds 3.0 µg-1” – Correct the unit! Check the format of reference!
Response: Thank you. We have corrected the unit and revised the reference format. Please check line now-60-61.
- The manuscript needs a careful perusal. Check the format of reference through the manuscript. Most frequently the year should be deleted, but somewhere the numbering is missing.
Response: Thank you. We have deleted the year format. Moreover, the missing numbering has been added.
Some other examples to correct:
- line 72: include
Response: Thanks. “Include” has been added. Please check line now-72.
- lines 135, 136: Mostafa et al., 2020. On other sites (lines 151, 153) Mostofa et al. 2020 is written. Check and correct it!
Response: We have checked and corrected it.
- Uniform the writing format of µL and mL. In some cases (e.g., in sections 2.8 and 2.11) „µl” is used.
Response: µL and mL has been added as uniform way throughout the manuscript. Please check.
- line 552: (SA) mediated
Response: “(SA) mediated” has been added. Please check line now-564.
- line 565: SPM mediated
Response: We have used abbreviation “Spm” to make uniform throughout the manuscript instead of “SPM”. Please check line now-577.
- line 598: associated with
Response: The term associated with has been added in the line. Please check line now-611.
Reviewer 2 Report
Selenium (Se) pollution gives severe impacts on human health as well as agriculture. In this study the authors provide experimental evidence that the polyamine spermine (Spm) in combination with nitic oxide (NO) application protects wheat seedlings from the Se-induced phytotoxicity. I believe that the results presented in this manuscript are worthy of publication. Before its publication the authors are requested to consider the following points.
The authors demonstrated that an Spm-NO (SNP) combination protects the Se-induced toxicity in wheat seedlings. The results presented all appear to support the authors statements. However, similar conclusions can be found in the previous studies using different plant species or different heavy metals. Although I am aware of the importance of this study in agriculture, I must say that novelty of study is weak. It would be nice if the authors could emphasize wheat specific characteristics or possible application of the findings to practical agriculture.
1. Compost
Polyamines can be produced in a process of protein degradation in soils by bacterial activities. Compost includes degraded proteins which could be a source of polyamines in the field. I wonder why the authors used compost in the mixture of clay soil (5:1, v:v) to investigate polyamine effects. Although it seems that compost was sterilized, plant culture in an ambient condition allows bacterial growth in the clay soils. To avoid this, it is recommended to use inorganic substances such as Perlite or Vermiculite rather than compost. To evaluate the results shown in this study, more information of the compost used in this study is necessary, at least.
2. How to apply chemicals
It is very difficult to follow how to apply Se, SNP and cPTIO to the plants. It seems that solutions of those chemicals were sprayed on leaves. Although their final concentrations are described in the manuscript, I cannot find their amounts (volume). Describe more carefully about the treatments in Materials and Methods. It is also difficult to follow the rationale of the treatments of leaves. As the authors stated in Introduction, Se toxicity is primarily attributed to Se-polluted soils. It would be helpful if the authors mention the rationale of experimental design of this study in Materials and Methods.
3. Griess method
This study applied the Griess method to quantify NO. The method measuring the NO oxidation product nitrite has been used for quantifying NO in animal tissues. Since plant cells include abundant nitrate (NO3-) and nitrite (NO2-), the Griess method has been not recommended to measure NO in plants. Note that nitrite can be detectable in chloroplasts when photosynthesis is inhibited in light conditions. Please describe in Materials and Methods how to distinguish the NO oxidation product nitrite from the intermediate metabolite nitrite of the nitrate assimilation.
4. Chemistry
PAs carry a positive charge on each nitrogen atom at neutral pH and can interact with polyanionic molecules. It may absorb selenate ion (SeO42-) (Nishimura et al. 2007: doi.org/10.1080/01496390701513107). Since chemistry between Se, NO and other molecules are complex, it may be difficult to estimate its effective concentration in a whole plant culture system. To avoid such chemistry issues, it would be recommended to set an appropriate control. In most cases, the polyamine Spm or spermidine (Spd) is biologically effective whereas putrescine (Put) is ineffective. I would suggest the authors to verify the Spm effects with Put as a negative control in a representative experiment.
5. Model illustration
I would suggest the authors to reconsider the design of the Figure 9. This illustration may lead to misunderstanding for the non-specialists.
a) Draw the cells of leaves and roots separately
b) Illustrate the localization of each event
c) Include the final consequence such as oxidative damage, growth inhibition, cell death etc.
If the authors intend to show only a conceptional scheme, remove colored boxes. If the authors intend to argue photosynthesis, an illustration needs to include differences in day and night. Note that most of reducing power as well as ATP requires light energy for plants. Thus, the physiological responses in plants should be separately considered in day/night metabolism.
The “e.g.” along with many abbreviated terms can be removed from the illustration for simplification.
There is no explanation for the difference in colors for the boxes and lines. Use of green and red colors should be carful for color blind readers.
Author Response
Selenium (Se) pollution gives severe impacts on human health as well as agriculture. In this study the authors provide experimental evidence that the polyamine spermine (Spm) in combination with nitic oxide (NO) application protects wheat seedlings from the Se-induced phytotoxicity. I believe that the results presented in this manuscript are worthy of publication. Before its publication the authors are requested to consider the following points.
The authors demonstrated that an Spm-NO (SNP) combination protects the Se-induced toxicity in wheat seedlings. The results presented all appear to support the authors statements. However, similar conclusions can be found in the previous studies using different plant species or different heavy metals. Although I am aware of the importance of this study in agriculture, I must say that novelty of study is weak. It would be nice if the authors could emphasize wheat specific characteristics or possible application of the findings to practical agriculture.
Response: We appreciate the reviewer comments and although there are some similar previous data, we believe that any specific potential application should be done with plants with agronomical relevance as it is wheat. And this work goes in this direction.
- Compost
Polyamines can be produced in a process of protein degradation in soils by bacterial activities. Compost includes degraded proteins which could be a source of polyamines in the field. I wonder why the authors used compost in the mixture of clay soil (5:1, v:v) to investigate polyamine effects. Although it seems that compost was sterilized, plant culture in an ambient condition allows bacterial growth in the clay soils. To avoid this, it is recommended to use inorganic substances such as Perlite or Vermiculite rather than compost. To evaluate the results shown in this study, more information of the compost used in this study is necessary, at least.
Response: Thank you for your comments. To sterilize the soil and compost is a standard procedure to eliminate undesirable microorganisms which could interfere in the analyses but we cannot discard any potential bacterial growth during the development of the experiments. In any case, we have added supplementary Table 1 with the soil and compost analysis in the revised manuscript.
- How to apply chemicals
It is very difficult to follow how to apply Se, SNP and cPTIO to the plants. It seems that solutions of those chemicals were sprayed on leaves. Although their final concentrations are described in the manuscript, I cannot find their amounts (volume). Describe more carefully about the treatments in Materials and Methods. It is also difficult to follow the rationale of the treatments of leaves. As the authors stated in Introduction, Se toxicity is primarily attributed to Se-polluted soils. It would be helpful if the authors mention the rationale of experimental design of this study in Materials and Methods.
Response: We have included a scheme to show the experimental design (new Figure 1). For each treatment, the applied volume was 30 mL.
- Griess method
This study applied the Griess method to quantify NO. The method measuring the NO oxidation product nitrite has been used for quantifying NO in animal tissues. Since plant cells include abundant nitrate (NO3-) and nitrite (NO2-), the Griess method has been not recommended to measure NO in plants. Note that nitrite can be detectable in chloroplasts when photosynthesis is inhibited in light conditions. Please describe in Materials and Methods how to distinguish the NO oxidation product nitrite from the intermediate metabolite nitrite of the nitrate assimilation.
Response: The reviewer is right that the Griess method determines the nitrite content as the final product of NO oxidation and we mentioned it the material and methods. It should be mentioned that the chemistry of the NO as a radical molecule in a cellular context is quite complex because it reacts with other radicals like superoxide dismutase to generate peroxynitrite or with peptides/proteins through posttranslational modifications (S-nitrosation or nitration). Therefore, this method should be considered an estimation of NO content in the assayed experimental conditions. Although there are other more specific techniques like ozone chemiluminescence or electron paramagnetic resonance (EPR) spectroscopy but they were not available in our laboratory.
- Chemistry
PAs carry a positive charge on each nitrogen atom at neutral pH and can interact with polyanionic molecules. It may absorb selenate ion (SeO42-) (Nishimura et al. 2007: doi.org/10.1080/01496390701513107). Since chemistry between Se, NO and other molecules are complex, it may be difficult to estimate its effective concentration in a whole plant culture system. To avoid such chemistry issues, it would be recommended to set an appropriate control. In most cases, the polyamine Spm or spermidine (Spd) is biologically effective whereas putrescine (Put) is ineffective. I would suggest the authors to verify the Spm effects with Put as a negative control in a representative experiment.
Response: Thank you for your recommendation. Nevertheless, to do this negative control will take to prepare a new set of experiments involving a minimum of 2 months. We believe that it is not strictly necessary in the present manuscript, although we will include it in our future studies.
- Model illustration
I would suggest the authors to reconsider the design of the Figure 9. This illustration may lead to misunderstanding for the non-specialists.
- a) Draw the cells of leaves and roots separately
- b) Illustrate the localization of each event
- c) Include the final consequence such as oxidative damage, growth inhibition, cell death etc.
If the authors intend to show only a conceptional scheme, remove colored boxes. If the authors intend to argue photosynthesis, an illustration needs to include differences in day and night. Note that most of reducing power as well as ATP requires light energy for plants. Thus, the physiological responses in plants should be separately considered in day/night metabolism.
The “e.g.” along with many abbreviated terms can be removed from the illustration for simplification.
There is no explanation for the difference in colors for the boxes and lines. Use of green and red colors should be carful for color blind readers.
Response: Many thanks for your recommendation and suggestions. We have tried our best to improve the model figure to show the insight mechanisms through the revised figure which is now Fig. 10. We hope that the reviewer finds these changes appropriate.
Round 2
Reviewer 2 Report
The manuscript has been improved by incorporating the comments and suggestions. I think that the revision of the Figure 10 helps the readers to follow the paper.